# The Influence of Mg-Impurities in Raw Materials on the Synthesis of Rankinite Clinker and the Strength of Mortar Hardening in CO_2_ Environment

**DOI:** 10.3390/ma16072930

**Published:** 2023-04-06

**Authors:** Raimundas Siauciunas, Edita Prichockiene, Zenonas Valancius

**Affiliations:** Department of Silicate Technology, Kaunas University of Technology, Radvilenu pl. 19, 50270 Kaunas, Lithuania

**Keywords:** rankinite, akermanite, Mg-containing impurities, curing in CO_2_ environment

## Abstract

The idea of this work is to reduce the negative effect of ordinary Portland cement (OPC) manufacture on the environment by decreasing clinker production temperature and developing an alternative rankinite binder that hardens in the CO_2_ atmosphere. The common OPC raw materials, limestone and mica clay, if they contain a higher MgO content, have been found to be unsuitable for the synthesis of CO_2_-curing low-lime binders. X-ray diffraction analysis (ex-situ and in-situ in the temperature range of 25–1150 °C) showed that akermanite Ca_2_Mg(Si_2_O_7_) begins to form at a temperature of 900 °C. According to Rietveld refinement, the interlayer distances of the resulting curve are more accurately described by the compound, which contains intercalated Fe^2+^ and Al^3+^ ions and has the chemical formula Ca_2_(MgO_0.495_·FeO_0.202_·AlO_0.303_)·(FeO_0.248_·AlO·Si_1.536_·O_7_). Stoichiometric calculations showed that FeO and Al_2_O_3_ have replaced about half of the MgO content in the akermanite structure. All this means that only ~4 wt% MgO content in the raw materials determines that ~60 wt% calcium magnesium silicates are formed in the synthesis product. Moreover, it was found that the formed akermanite practically does not react with CO_2_. Within 24 h of interaction with 99.9 wt% of CO_2_ gas (15 bar), the intensity of the akermanite peaks does not practically change at 25 °C; no changes are observed at 45 °C, either, which means that the chemical reaction does not take place. As a result, the compressive strength of the samples compressed from the synthesized product and CEN Standard sand EN 196-1 (1:3), and hardened at 15 bar CO_2_, 45 °C for 24 h, was only 14.45 MPa, while the analogous samples made from OPC clinker obtained from the same raw materials yielded 67.5 MPa.

## 1. Introduction

The ordinary Portland cement (OPC) industry is responsible for about 5–7 wt% of global anthropogenic greenhouse gas emissions. One of the most promising alternative binders is rankinite Ca_3_Si_2_O_7_: it is synthesized at a temperature lower than 200 °C while requiring almost twice less carbonate rock, and it is able to store a considerable amount of CO_2_ in the concrete structure.

Rankinite is a low-lime non-hydraulic calcium silicate (Ca_3_Si_2_O_7_ or C_3_S_2_). It was first found at Scawt Hill, Ireland, and named in honor of G. A. Rankin, a physical chemist at the Geophysical Laboratory, Washington, D.C., USA [1]. It is a very rare mineral that can be found in paralava rocks [2]. Natural rankinite only slightly varies in composition from pure Ca_3_Si_2_O_7_ because ions of other metals hardly ever incorporate into its crystal lattice structure. Rankinite belongs to the silicates of the sorosilicate group (Si_2_O_7_), in which a central O connects two SiO_3_ units [3]. The interatomic distances between Ca and O show a range of values from 2.25 to 2.90 Å, while the angles O-Ca-O are between 60 and 87 degrees [4]. The rankinite structure is ordered in successive layers of Ca atoms, and Si_2_O_7_ groups parallel to the *z* direction, with the Si_2_O_7_ units oriented towards the *z* direction, which results in the lower compressibility of the mineral in this direction [3]. 

Rankinite can be produced from the same calcareous and siliceous raw materials as OPC clinker, and its manufacture needs neither specialized equipment nor additional unit operations as already existing OPC plants can be used without virtually any modification. The rankinite calcination temperature is about 200–250 °C [5,6,7,8] lower than that of OPC. The non-hydraulic nature of rankinite eliminates the usage of gypsum as a set-controlling component. Moreover, the carbonation curing of rankinite is a relatively speedy process. The process can be carried out under ambient and elevated pressure, at a temperature of 20–60 °C, whereas the duration is usually up to 24 h. It is far more attractive and efficient than the conventional 28-day curing cycle of OPC concretes. Above all, rankinite carbonation opens the possibility for the permanent sequestration of CO_2_ in a hardened concrete structure.

B. Qian et al. [9] determined the optimal sintering temperature and duration of C_3_S_2,_ which is approximately 1300 °C and 4 h, respectively. The materials used for the synthesis were industrial lime and quartzite. The authors claim that, at temperatures above 1300 °C, the formation of rankinite no longer occurred. With a prolonged sintering time, C_3_S_2_ formed via the consumption of β-C_2_S and γ-C_2_S.

K. Wang et al. [10] synthesized pure rankinite by calcining a C-S-H gel precursor at a temperature of 1300 °C for 2 h. The C-S-H gel was prepared hydrothermally at 60 °C for 6 h from a mixture with a molar ratio of CaO/SiO_2_ = 1.5. H. Zwang et al. [11] prepared C_3_S_2_ from the mixture of CaCO_3_ and SiO_2_ (at 3:2 molar ratios) that was calcined at a temperature of 1440 °C for 2 h. B. Lu et al. [12] obtained a low calcium clinker containing rankinite by sintering the stoichiometric mixture of calcium oxide, amorphous silica, aluminum oxide, and ferroferric oxide. The mixtures were heated to the target temperature of 1250–1320 °C for 2 h. The author determined the sintering temperature for this type of clinker was about 1320 °C and included mostly C_3_S_2_, γ-C_2_S, and little C_2_AS. G. Hou et al. [13] synthesized a new type of clinker with a self-pulverization effect from industrial raw materials (limestone, sandstone, and clay) at a temperature of 1260–1320 °C. It mainly consisted of rankinite and γ-dicalcium silicate. The main advantage is that clinker is obtained in the form of powder, which significantly reduces energy consumption during grinding.

Rankinite, as outlined above, can be synthesized from the same raw materials as OPC clinker. However, researchers seek to use other, and usually non-traditional, materials, waste, or by-products from other productions. W. Ashraf and J. Olek [14] synthesized rankinite form CaO and fumed silica with CaO/SiO_2_ molar ratio 3:2 at the heating temperature of 1250 °C. A. Smigelskyte et al. obtained this compound from a silica–calcite sedimentary rock (opoka) also at 1250 °C [15]. The low CO_2_ products offered by Solidia Technologies (USA) are probably the most widely known today. They produce non-hydraulic cement, which main components are wollastonite and rankinite [16]. Due to the lower amount of CaO present in this binder, the production temperature is only 1200 °C. When assessing all stages of production, the technology suggested by *Solidia* can reduce the carbon footprint by up to 70 wt% [17]. However, it should be noted that despite the fact that the synthesis temperature of rankinite binder is ~200 °C lower than that of OPC clinker, it still remains quite high—at about 1250 °C.

One of the ways to avoid the calcination step is to use hydrothermally synthesized calcium silicate hydrates as a binder instead of anhydrous calcium silicates in the production of mortars that harden in the CO environment. It was found that, from α-C_2_SH/kilchoanite and sand mixture (1:3 by mass), it is possible to obtain products with a compressive strength of ~25 MPa [18,19]. Another way is to use industrial solid wastes, such as steel slag, mine tailings, fly ash, cement kiln dust, and others [20]. They contain intrinsic alkalinity (e.g., Ca, Mg, Al, Si, and alkali), which may lead to a reaction with CO_2_. The higher the basicity of such substances, the more CO_2_ is combined [21,22,23,24]. The main problem preventing the wider use of these materials is that most of the Ca and Mg is not present in the pure form but is combined with silicates or another complex oxide phase [20]. Moreover, in natural rocks, the impurities usually alter the course of hydrothermal and high-temperature sintering reactions along with the mineral composition of the resulting products; hence, their effects still need to be studied in detail.

However, rankinite synthesis from natural rocks still requires careful and thorough research because the impurities contained in the rocks can significantly influence the course of high-temperature reactions and the mineralogical composition of the resulting products.

One such impurity is Mg-containing compounds. Reactive MgO is produced at a low calcination temperature (700–900 °C) and offers a high CO_2_ binding capacity [25]. It can also be obtained from alternative sources, such as seawater or rejected brine [26,27]. The binder that hardens in a CO_2_ environment is obtained in two stages—firstly, MgO is hydrated to form Mg(OH)_2_, and its further carbonization leads to the formation of a range of hydrated magnesium carbonates [20]. Other suitable materials for mineral carbonation are Mg silicates, such as forsterite Mg_2_SiO_4_ or serpentine Mg_6_Si_4_O_10_(OH)_8_ [28]. It should be noted that Mg silicates are abundant in the Earth’s crust and can be used particularly successfully in the regions where limestone and clay are scarce [29]. Magnesium silicates can be either used as a permanent CO_2_ sink or as a raw material to produce MgO [30]. The magnesium oxides derived from magnesium silicates (MOMS) approach is based on magnesium silicate raw materials, which have the advantage, relative to limestone, of containing no chemically bound (“fossil”) CO_2_. This means that, unlike the other approaches, MOMS, at least in theory, provides the possibility of making concretes with a significantly negative carbon footprint, especially if carbonation hardening is used. Thus, MgO-based binders offer the possibility to produce concretes and mortars with a reduced carbon footprint [31].

Mg-containing impurities (usually dolomite) are often present in limestone and mica clay—the most commonly used raw materials for OPC production worldwide. It does not affect the formation of alite and belite (1450 °C). The current data suggest that when synthesizing rankinite or wollastonite (1100–1250 °C), Mg compounds form bredigite and akermanite [32,33], but their influence on the synthesis of low-lime calcium silicates is not completely clear. The aim of this work is to determine the influence of Mg-impurities present in the raw materials on the synthesis of a rankinite-based binder and on the strength properties of mortars hardened in a CO_2_ environment.

## 2. Materials and Methods

### 2.1. Characterization of the Raw Materials

Limestone from Karpenai deposit (Lithuania) was dried for 24 h at a temperature of 100 ± 1 °C and ground in a ball mill to a specific surface area by Blaine S*_a_* = 547 m^2^/kg with ignition losses of 43.26 wt%. The chemical composition is given in Table 1. CaO is the dominant compound in the raw material (47.95 wt%). The amount of MgO, which is included in the composition of limestone impurity, dolomite, is moderate and equal to almost 4 wt%. The number of other oxides does not exceed 1 wt%.

Clay of the Saltiskiai deposit was dried for 24 h at 100 ± 1 °C and ground to S*_a_* = 614 m^2^/kg; ignition losses—14.9 wt%. The chemical composition is given in Table 1.

OPC clinker granules were taken from *JSC Akmenes cementas* and ground in a ball mill to S*_a_* = 323 m^2^/kg in 18 h. Its chemical composition is given in Table 1.

CEN Standard sand EN 196-1 (Normsand, Beckum, Germany).

### 2.2. Experimental Methods

#### 2.2.1. Samples Preparation, Sintering, and Solidification

Two mixtures were prepared from Karpenai limestone and Saltiskiai clay (their composition and properties are fairly close to those most widely used in the world for the production of OPC), with molar ratios of CaO/SiO_2_ = 1.5 and 1.75. The first ratio corresponds to the stoichiometry of rankinite, whereas, in the second case, the amount of CaO is increased due to the fact that, even before the beginning of the reactions of calcium silicate formation, part of CaO is consumed for the synthesis of calcium magnesium silicates, i.e., the molar ratio of CaO/SiO_2_ decreases in the reacting mixture. The necessary amounts of components were weighed and then poured into sealed plastic containers; then, 6 porcelain grinding bodies were placed into them (to ensure the quality of homogenization), and the content was mixed for 2 h at a speed of 48 rpm in a homogenizer Turbula Type T2F (Muttenz, Switzerland) for 1 h at 49 rpm.

The dry mixtures were mixed with ~20 wt% distilled water, homogenized, and manually formed into pellets with a diameter of ~15 mm. They were dried to a constant mass at a temperature of 100 ± 1 °C in a Kambič *S-25C* dryer (Kambič d.o.o., Semič, Slovenia). They were stored in sealed plastic containers and calcinated in air environment at temperatures of 1100 °C, 1150 °C, 1200 °C, and 1250 °C in the furnace Nabertherm LHT 08/16 (Nabertherm GmbH, Bremen, Germany), isothermal holding time 45 min. The fired pellets were crushed in an agate mortar to ~3 mm size grains, which were ground at a speed of 950 rpm in 6 min. In a vibrating cup mill Fritsch Pulverisette 9 (Fritsch GmbH, Idar-Oberstein, Germany). In preparation for instrumental analysis, the powders were sieved through a sieve with a mesh size of 80 µm.

CO_2_ curing samples were formed (1.0 kN/s speed) and compressed (1.5 kN/s) according to EN 196-1 and EN 12390-6 Standards while using the universal testing machine FORM + TEST MEGA 10-400-50 (Seider&Co GmbH, Riedlingen, Germany). During the preparation of the samples, the binding material fired at 1150 °C (CaO/SiO_2_ = 1.5) and OPC clinker at a mass ratio of 1:3 were dry mixed with CEN Standard sand EN 196-1. After that, the mixtures were moistened so that the amount of water and the binder would correspond to the ratio W/C = 0.3, thoroughly mixed again in a porcelain dish; then, cylinders sized Ø36 × 36 ± 1 mm were formed with a force of 12.5 kN with a hold of 20 s. The formed samples were weighed (to estimate the mass change after carbonization) and immediately placed in an autoclave (to prevent drying).

The carbonization process was carried out in a Parr Instruments pressure reactor, model 4555 (Parr Instrument Company, Moline, IL, USA), with a maximum working pressure of 131 bar, a volume of 18.75 L, and a temperature range of −10 to +350 °C (the system diagram is presented in Figure 1a). Before solidification, the autoclave was filled twice with CO_2_ gas to 2 bar, and immediately depressurized to atmospheric pressure so as to eliminate the presence of air. During the experiment, the reactor was filled with 99.9 wt% CO_2_ gas at a rate of 2.5 bar/min to the required value. At the end of the set holding time, the gas was released at the same rate. The samples were cured at temperatures from 25 to 45 °C under a pressure of 15 bar of carbon dioxide, and the holding time was from 4 h to 24 h. After curing, the samples (Figure 1b) were weighed, and while they were still wet, the compressive strength was determined. At least 4 samples were tested in each experiment. A specimen weighing ~10 g was taken from one sample of each series, dried at a temperature of 100 ± 1 °C to a constant mass, crushed in an agate mortar, sieved through a sieve with a mesh size of 80 µm, and analyzed by employing methods of instrumental analysis.

#### 2.2.2. Instrumental Analysis

The chemical composition analysis of samples was performed by X-ray fluorescence spectroscopy (XRF) on a Bruker X-ray S8 Tiger WD (Bruker AXS GmbH, Karlsruhe, Germany) spectrometer equipped with Rh tube with energy of up to 60 keV. The materials were ground, put through a sieve with an 80-μm mesh, and compressed into tablets of 40 mm with a force of 20 kN. Samples were measured in helium atmosphere, and data were analyzed with SPECTRA Plus V.2 QUANT EXPRESS standard-less software.

The X-ray diffraction analysis (XRD) was performed on the D8 Advance diffractometer (Bruker AXS GmbH, Karlsruhe, Germany) operating at the tube voltage of 40 kV and tube current of 40 mA. The X-ray beam was filtered with Ni 0.02 mm filter to select the CuKα wavelength. Diffraction patterns were recorded in a Bragg-Brentano geometry using a fast counting detector Bruker Lynx Eye (Bruker AXS GmbH, Karlsruhe, Germany) based on a silicon strip technology. The samples were scanned over the range 2θ = 3–70° at a scanning speed of 6° min^−1^ using a coupled two theta/theta scan type. The software Diffrac.Eva v3.0 (Bruker AXS GmbH, Karlsruhe, Germany) and PDF-4 database were used for compound identification.

The thermal stability and phase transformation were determined by in-situ XRD analysis. It was carried out with a high-temperature camera MTC–hightemp (Bruker AXS, Karlsruhe, Germany) at a 25–1000 °C temperature range. The measurement was performed with a step width of 0.02 2θ and 0.5 s step^−1^ at a heating rate of 50 °C min^−1^ after equilibration for 5 min at the desired temperature.

The quantitative content of the crystalline phases was refined by the Rietveld method, as implemented in the TOPAS 4.2 software (Bruker AXS GmbH, Karlsruhe, Germany). The quantitative phase analysis (QPA) was carried out on samples after grinding by hand to pass a 40 μm sieve, ensuring an isotropic distribution of the crystals in the sample. For the quantitative determination of the amorphous phase, 20% of the internal standard (ZnO) was added to the sample. Additionally, the sample holder was rotated during the data recording to improve the particle statistics and obtain high-quality QPA data. The crystal structures used in the refinements were adopted from PDF 2022 database.

Thermal analyzer Linseis STA PT1000 (Linseis Massgeraete GmbH, Selb, Germany) was applied to simultaneous thermal analysis (STA; differential scanning calorimetric DSC, and thermogravimetry TG) studies. The heating was carried out under an N_2_ atmosphere at a heating rate of 10 °C·min^−1^; the temperature ranged from 40 up to 1000 °C. The ceramic sample handlers were used for compound identification.

The granulometric composition of the raw materials was determined by a laser particle size analyzer CILAS 1090 LD (Cilas, Orléans, France) with a measurement range of 0.04–500 µm. The dry method was used with an air supply of 6 bar.

The specific surface area of the powders was determined by the Blaine method using electronic air permeability apparatus CE091 (Toni Technik Baustoff GmbH, Berlin, Germany).

The density of the materials was determined with an *Ultrapyc 1200e pycnometer* (Quantachrome Instruments, Boynton Beach, FL, USA).

## 3. Results and Discussion

JSC *Akmenes Cementas* has been successfully producing *ordinary Portland cement* (OPC) from local raw materials (Karpenai deposit limestone and Saltiskiai deposit clay, Lithuania) for more than 7 decades. Its main constituents are also anhydrous calcium silicates—alite (3CaO·SiO_2_; ~66 wt%) and belite (2CaO·SiO_2_; ~10 wt%). The only significant difference is that the molar ratio CaO/SiO_2_ of the raw material mixture is by one and a half points higher (2.7) than that required for the synthesis of rankinite (1.5). It is likely that these raw materials are also suitable for the production of low-lime binders, which would harden in a CO_2_ environment. For comparison, we also determined the compressive strength and the mineralogical composition of mortar samples formed from OPC clinker produced by JSC *Akmenes Cementas*/sand mixture and hardened in a CO_2_ environment.

X-ray diffraction analysis (Figure 2a) shows that the limestone is dominated by calcite CaCO_3_ (PDF No. 00-001-0837). In addition, the raw material also contains dolomite CaMg(CO_3_)_2_ (PDF No. 01-071-4892) and a small amount of quartz (PDF No. 00-003-0419).

The simultaneous thermal analysis (STA) of limestone (Figure 2b) showed that, in the differential scanning calorimetry (DSC) curve, at 50–120 °C, there is a slight endothermic band, and the mass of the sample slightly decreases due to the removal of water residues. The decomposition of CaCO_3_ takes place at 810 °C. The mass loss in the temperature range of 730–820 °C is 41.97 wt% (Figure 2b; thermogravimetry TG curve). After recalculations, we see that the raw material contains 95.4 wt% (CaCO_3_ + MgCO_3_) or 53.4 wt% (CaO + MgO). This correlates well with the results of the chemical analysis (Table 1).

According to the curves of the granulometric analysis of the limestone (Figure 3a), it was determined that it contains 10 wt% of particles with a diameter of up to 1.10 µm, 50 wt% with a diameter of up to 5.16 µm, and 90 wt% with a diameter of up to 33.07 µm, while the average diameter of all particles is 11.47 µm. Moreover, the specific surface area was equal to 525 m^2^/kg.

Based on the granulometric analysis curves of Saltiskiai clay (Figure 3b), it was found that it contains 10 wt% of particles with a diameter of up to 0.86 µm, 50 wt%—up to 3.56 µm and 90 wt%—up to 21.92 µm, and the mean diameter of all particles is 7.89 µm.

X-ray diffraction analysis (Figure 4a) shows that quartz, in the form of sand, prevails in Saltiskiai clay. Among the clay minerals, the peaks of the highest intensity are characteristic of the minerals of the mica group—illite [K,H_3_O)Al_2_Si_3_AlO_10_(OH)_2_] (PDF No. 00-026-0911). There are also small peaks of montmorillonite [Na,Ca)_0.33_(Al,Mg)_2_(Si_4_O_10_)(OH)_2_·*n*H_2_O] (PDF No. 00-026-0911), muscovite [Kal_2_(AlSi_3_O_10_)(F,OH)_2_] (PDF No. 00-019-0814) and kaolinite [Al_2_Si_2_O_5_(OH)_4_] (PDF No. 00-03-0052). Calcite and dolomite, i.e., carbonates, were found. They are useful as a partial source of CaO for the synthesis of rankinite. STA results (Figure 4b) indicate the characteristic features of mica carbonaceous clays. The following three main peaks were observed: (1) the loss of the absorbed water (up to 150 °C), (2) organic matter burning (300–450 °C), and (3) the decomposition of dolomite and calcite (650–750 °C) [34]. The mass loss within the temperature range of 450–600 °C is 1.46 wt%. This is most likely due to the fact that Saltiskiai clay consists of a low amount of kaolinite and montmorilonite.

The mineralogical, as well as the granulometric composition of the OPC clinker, is presented in Figure 5.

In order to preliminarily determine the optimal calcination conditions for this mixture, high-temperature X-ray diffraction analysis was performed in the temperature range of 25–1150 °C. The data are presented in Table 2 and in Figure 6.

In the temperature range of 25–625 °C, new compounds are not formed—that is, calcite, quartz, dolomite and illite remain in the raw materials. New compounds are not formed in the temperature range of 25–625 °C; only calcite, quartz, dolomite, and illite remain in the raw materials. Meanwhile, at 650 °C—dolomite and, at 700 °C—illite starts to decompose (Figure 6). Akermanite (PDF No. 00-002-0824) begins to form at 700 °C. The latter compound and quartz do not fully react up to the maximum temperature of 1150 °C. Moreover, at the temperature of 925 °C, the formation of calcium silicate, larnite 2CaO·SiO_2_, begins. While the intensity of the peaks of larnite essentially does not change throughout the studied temperature range, ones of akermanite increase significantly. This means that a higher amount of Ca^2+^, Si^4+^, and Mg^2+^ ions are bound to the composition of this compound. The target mineral, rankinite, starts to form only at 1100 °C. When the temperature is increased to 1150 °C, the peak intensity of this compound increases only slightly. Based on the high-temperature XRD data, it was decided to burn the binder in the temperature range of 1100–1250 °C. The data are presented in Figure 7.

The obtained results are that then good agreement with the high-temperature XRD data. In the mixture with CaO/SiO_2_ = 1.5, the predominant compound is akermanite. Only traces of larnite and rankinite are identified. Since the duration of isothermal curing was 45 min (in contrast to the short retention time in high-temperature XRD), quartz had the time to fully react already at 1100 °C. The maximum firing temperature is 1200 °C because Saltiskiai clay contains alkaline compounds (K_2_O = 3.23 wt%), and the granules begin to melt and stick together at ~1250 °C.

An attempt to increase the molar ratio of the mixture to CaO/SiO_2_ = 1.75 did not work. Although the intensity of the akermanite peaks is significantly reduced, another calcium magnesium silicate is formed—bredigite Ca_7_Mg(SiO_4_)_4_ (PDF No. 00-035-0260). The quantitative oxide composition of akermanite is 41.14 wt% CaO, 14.78 wt% MgO, and 44.08 wt% SiO_2_. This means that 1 g of MgO binds almost 3 g of CaO and SiO_2_ each, and, in the reacting mixture, the CaO/SiO_2_ molar ratio hardly changes at all—the conditions for the formation of rankinite remain. Bredigite contains 5.99 wt% MgO, 58.31 wt% CaO and 35.70 wt% SiO_2_. Thus, 1 g of MgO binds a significantly higher amount of CaO (9.73 g) than SiO_2_ (5.96 g)—the stoichiometry of the mixture is changed, and rankinite is no longer formed.

It was decided to investigate the influence of the sintering duration on the rankinite formation processes. The mixture with CaO/SiO_2_ = 1.5 was fired for 30 and 60 min at 1150 and 1200 °C.The data are presented in Figure 8.

The obtained results showed a similar tendency, as follows: akermanite is still the predominant compound, while smaller amounts of rankinite and larnite were formed.

When the conversion of the composition of the raw material mixture from moles to percentages was applied, it was found that it contained 58.33 wt% of CaO and 41.67 wt% of SiO_2_. Most of the CaO was added with limestone (which contains 3.94 wt% MgO) and SiO_2_—with clay (which contains 4.0 wt% MgO). Arithmetically, it is easy to calculate that 3.97 wt% MgO is added together with the necessary amounts of CaO and SiO_2_. As mentioned above, akermanite contains 41.14 wt% CaO, 14.78 wt% MgO, and 44.08 wt% SiO_2_, thus, the following:14.78 wt% MgO → 100 wt% akermanite
3.97 wt% MgO → x wt% akermanite  x = 26.86 wt%

This value is very close to the theoretical amount of akermanite in the mixture calculated from the molar masses, as follows:$${\mathrm{Ca}}_{2}\mathrm{Mg}\left[{\mathrm{SiO}}_{7}\right]=\frac{2\mathrm{CaO}}{112}\cdot \frac{\mathrm{MgO}}{40}\cdot \frac{2{\mathrm{SiO}}_{2}}{120};\;\mathrm{molar}\;\mathrm{mass}=272$$
MMgO = 40 → Makermanite = 272
3.97 wt% MgO → xwt% akermanite x = 26.99 wt%

Unfortunately, according to the XRD data, it appears that the mixture contains a significantly higher quantity of akermanite. The orientation in 2 1 1 plane (*d* –0.2866 nm) was used. Reliability values: R_exp_—2.61; R_wp_—5.67; GOF—2.17. This would not be surprising since it is well known that additional divalent and trivalent metal ions can be inserted into the crystal lattice structure of this compound, thus increasing the amount of akermanite in the synthesis product. In order to clarify this and determine the exact quantitative composition of the binder, a Rietveld refinement analysis was performed. The original data are shown in Figure 9, and its graphical representation is provided in Figure 10.

After carrying out the Rietveld refinement, we noticed that the interlayer distances of the resulting curve are more accurately described by another standard, PDF No. 01-082-3036. It contains intercalated Fe^2+^ and Al^3+^ ions, the chemical formula is Ca_2_(MgO_0.495_·FeO_0.202_·AlO_0.303_)·(FeO_0.248_·AlO·Si_1.536_·O_7_), and a molecular weight equal to 286.45. It follows that:M_MgO_ = 40 × 0.495 = 19.8 → M_akermanite (Fe, Al)_ = 286.45
3.97 wt% MgO → x wt% akermanite_(Fe,Al)_  x = 57.43 wt%

These data allow us to state that only ~4 wt% of MgO in the raw material determines that ~60 wt% of calcium magnesium silicates are formed in the synthesis product. Accordingly, the amount of calcium silicates—which give the samples mechanical strength during carbonization in a CO_2_ environment—is low (rankinite—only ~19 wt%, larnite—~15 wt%). These results correlate well with the chemical composition of Saltiskiai clay as it contains 14.0 wt% Al_2_O_3_ and 7.04 wt% Fe_2_O_3_ (Table 1). No other compounds containing Al_2_O_3_ or Fe_2_O_3_ were identified in the synthesis products; hence, it is logical to assume that Al^3+^ and Fe^3+^ ions are isomorphically intercalated into the lattice structure of akermanite crystals.

The fundamental question remains—whether akermanite reacts with CO_2_ and ensures the sufficient compressive strength of mortars. In order to determine this, from a stoichiometric mixture of analytical grade reagents (CaO, SiO_2_, MgO) at a temperature of 1150 °C for 1 h, a product was synthesized, in which akermanite was the predominant compound (Figure 11, pattern 1). It is true that, under these conditions, some of the initial components do not react, but this was not the main purpose of the study. Our key objective was to determine whether akermanite interacts with CO_2_. Unfortunately, it can be concluded that very slowly. Within 24 h interaction of samples (Ø36 × 10 ± 1 mm, dry powder mixed with 10 wt% of water and compacted with a pressure of 12.5 kN) with 99.9 wt% CO_2_ gas, neither at 25 °C (Figure 11, pattern 2) nor at 45 °C (Figure 11, pattern 3), the intensity of the akermanite peaks were similar (Figure 12). It means that this compound reacts very difficult with CO_2_ gas.

To confirm this, the Net area of the main non-overlapping peak of akermanite (*d* = 0.4069 nm) was calculated. For this purpose, Diffrac.Eva software was used. The calculations were performed on an interval between two points, called the left angle (21.397°) and right angle (22.193°). The mentioned angles of the scan point are the closest to the entry points. These statistical calculations provide information about the position of the peak maximum and its net area. It should be mentioned that instrumental broadening was not considered.

The calculations confirmed that during the carbonization of the akermanite sample, its peak profile, and intensity only slightly changed in the XRD curves. In the product synthesized at a temperature of 1150 °C, the area of the akermanite peak (*d* = 2.810 nm) is 0.9335 relative units. After treatment with 99.9 wt% CO_2_ (15 bar) for 24 h at a temperature of 25 °C, it decreases to 0.8705 r.u. and at a temperature of 45 °C—to 0.7064 r.u.

In order to finally ascertain the suitability of limestone-clay raw materials with a moderate amount of Mg-containing impurities for the production of binders that harden in a CO_2_ environment, the product was burnt from Karpenai limestone and Saltiskiai clay (CaO/SiO_2_ = 1.5, 1150 °C in 45 min; Figure 7, curve 2) and mixed with the standard sand (mass ratio 1:3, water/binder ratio 0.25). From this mixture, Ø36 × 36 ± 1 mm samples—cylinders were formed and cured for 24 h in an atmosphere of 15 bar 99.9 wt% CO_2_ at temperatures 25 and 45 °C. To compare the results, analogous samples were formed and hardened under the same conditions from OPC clinker produced by JSC *Akmenes cementas* (its mineralogical composition is presented in Figure 12a) and standard sand. Figure 12b shows the XRD pattern of the carbonized sample (1 part clinker and 3 parts sand).

The XRD pattern of OPC clinker (Figure 12a) shows all the following mineral phases characteristic of this type of binder: alite (PDF No. 00-042-0551), belite (PDF No. 00-029-0369), tricalcium alumina (PDF No. 00-038-1429), calcium aluminum iron oxide (PDF No. 04-006-8316), periclase (PDF No. 01-076-3013), and sodium calcium aluminum oxide (PDF No. 00-026-0957). After carbonization, a lot of quartz remains in the sample because it does not react with CO_2_ gas (Figure 12b). As expected, most of alite and all of the belite, calcium aluminates, and calcium aluminum ferrites completely reacted during carbonization, thus, forming calcite and a SiO_2_/Al_2_O_3_-rich gel. The obtained calcite densifies the structure of the samples and ensures their good physical and mechanical properties. The obtained compressive strength data are presented in Table 3.

We would like to mention that after synthesizing rankinite from limestone and opoka (a sedimentary lime-silica rock; it contains only 0.61 wt% MgO), forming and curing samples from a mixture of it and sand (the conditions were the same as in this work), their compressive strength exceeded 45 MPa [32]. Since the compressive strength of the samples differs by 5–6 times, the conclusion is unequivocal—if the raw materials contain a certain number of Mg-containing impurities (in this case—limestone and easily fusible mica clay), they are not suitable for the production of low-lime binders that harden in CO_2_ environment.

## 4. Conclusions

Synthesis of low-lime calcium silicates from natural rocks requires careful and thorough research because the impurities contained in them can significantly influence the course of high-temperature reactions and the mineralogical composition of the resulting products;The impurity of Mg-containing compounds in limestone and mica clay, the most widely used raw materials for ordinary Portland cement production in the world, may determine that they are unsuitable for the synthesis of calcium silicate, rankinite, which hardens in a CO_2_ environment. Already at a temperature of 900 °C, akermanite with intercalated Fe^3+^ and Al^3+^ ions begins to form, which reacts very difficult with CO_2_ gas;Only ~4 wt% of the amount of MgO in the mixture of raw materials determines that more than 60 wt% of akermanite is formed in the synthesis product. As a result, the compressive strength of mortars hardened in a CO_2_ environment is six–seven times lower than the strength of OPC clinker samples made from the same raw materials and hardened under the same conditions.

## Figures and Tables

**Figure 1 materials-16-02930-f001:**
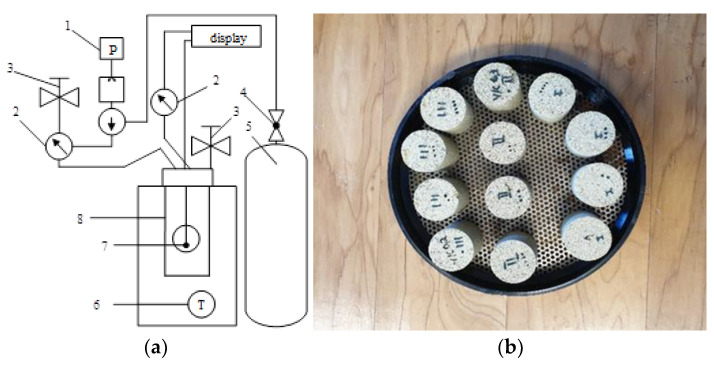
Technological system of samples hardening in CO_2_ environment (**a**) and hardened samples (**b**): 1—pneumatic pump; 2—manometers; 3—shutters; 4—ball valve; 5—CO_2_ gas cylinder; 6—temperature sensor; 7—autoclave; 8—heater-thermostat.

**Figure 2 materials-16-02930-f002:**
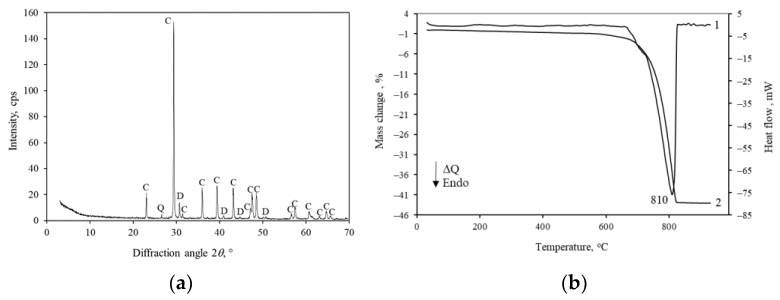
XRD pattern (**a**) and STA curves (**b**); 1—DSC, 2—TG of Karpenai limestone. Indexes: C—calcite, D—dolomite, Q—quartz.

**Figure 3 materials-16-02930-f003:**
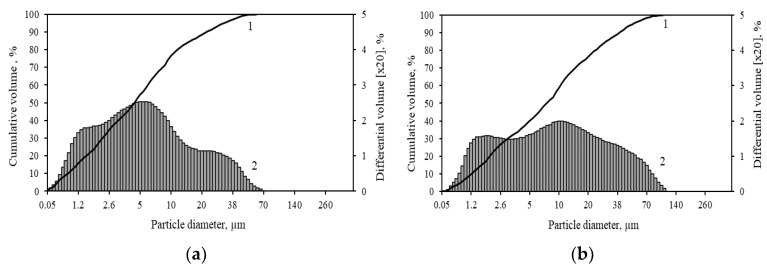
Particle size distribution of limestone (**a**) and clay (**b**), where 1—cumulative volume, 2—differential volume.

**Figure 4 materials-16-02930-f004:**
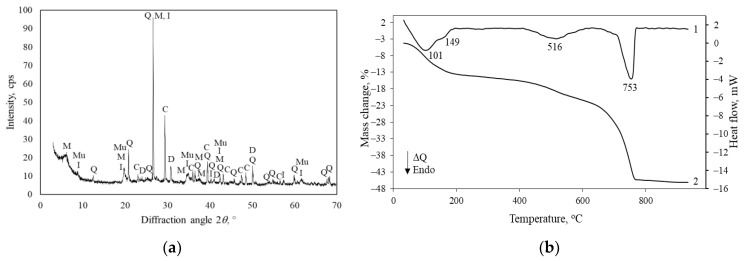
XRD pattern (**a**) and STA curves (**b**); 1—DSC, 2—TG of Saltiskiai clay. Indexes: C—calcite, D—dolomite, Q—quartz, I—illite, M—montmorillonite, Ms—muscovite.

**Figure 5 materials-16-02930-f005:**
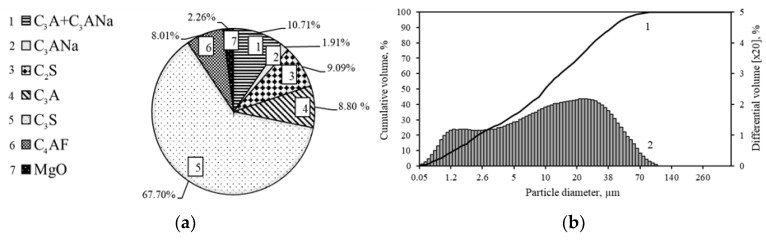
Mineralogical (**a**) and granulometric (**b**) composition of OPC clinker, where 1—cumulative volume, 2—differential volume. Indexes: C_3_S—alite, C_2_S—belite, C_3_A—calcium aluminum oxide, C_3_Ana—sodium calcium aluminum oxide, C_4_AF—calcium aluminum iron oxide, MgO—periclase.

**Figure 6 materials-16-02930-f006:**
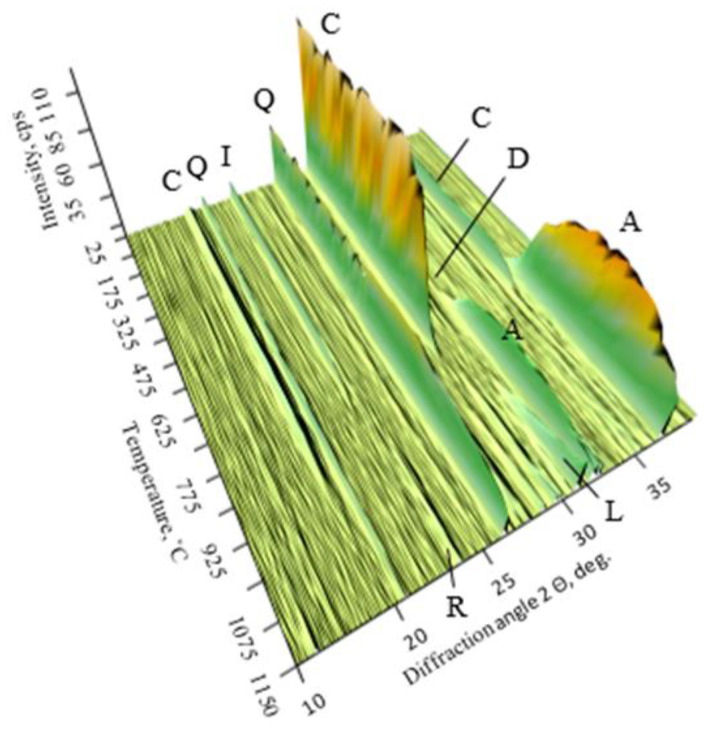
XRD diagram of the synthesis product from a mixture of Karpenai limestone and Saltiskiai clay (CaO/SiO_2_ = 1.5) in the temperature range of 25–1150 °C.

**Figure 7 materials-16-02930-f007:**
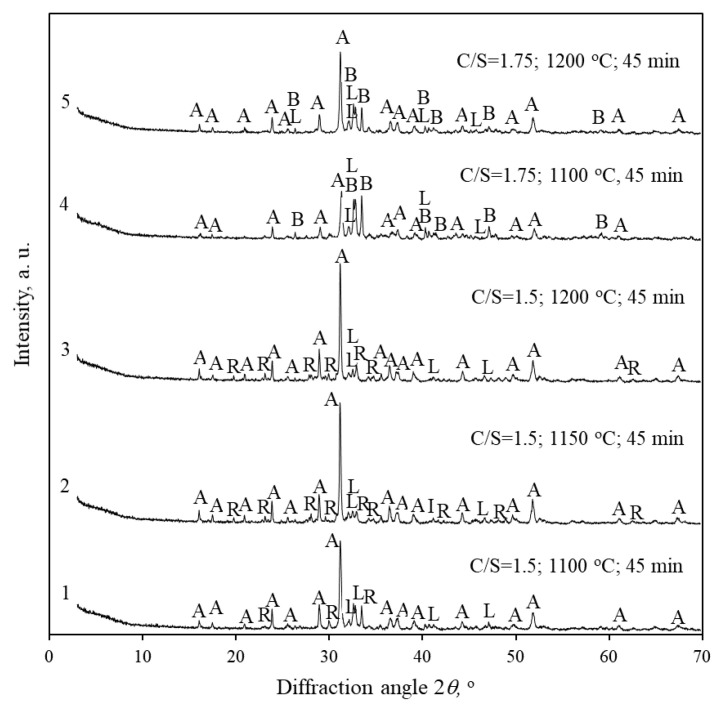
XRD patterns of Karpenai limestone—Saltiskiai clay mixtures fired at different temperatures. Indexes: A—akermanite, B—bredigite, R—rankinite, L—larnite.

**Figure 8 materials-16-02930-f008:**
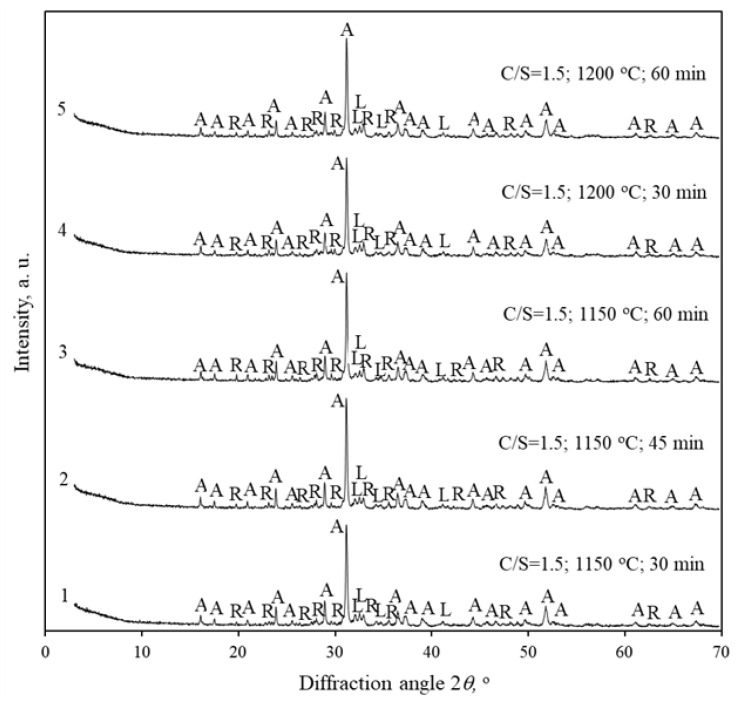
XRD patterns of Karpenai limestone—Saltiskiai clay mixture fired for different durations at 1150 and 1200 °C. Indexes: A—akermanite, R—rankinite, L—larnite.

**Figure 9 materials-16-02930-f009:**
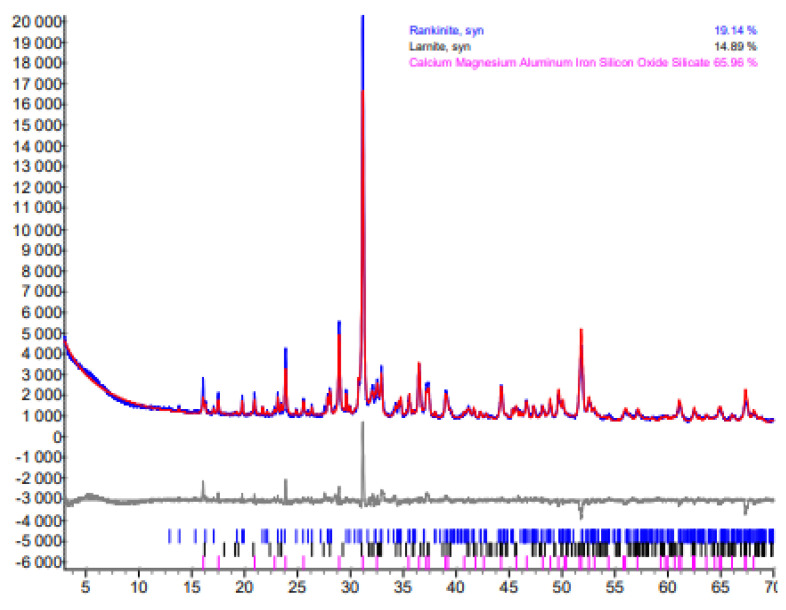
Rietveld analysis curve of the product synthesized at 1150 °C in 45 min from a mixture of Karpenai limestone and Saltiskiai clay with CaO/SiO_2_ = 1.5.

**Figure 10 materials-16-02930-f010:**
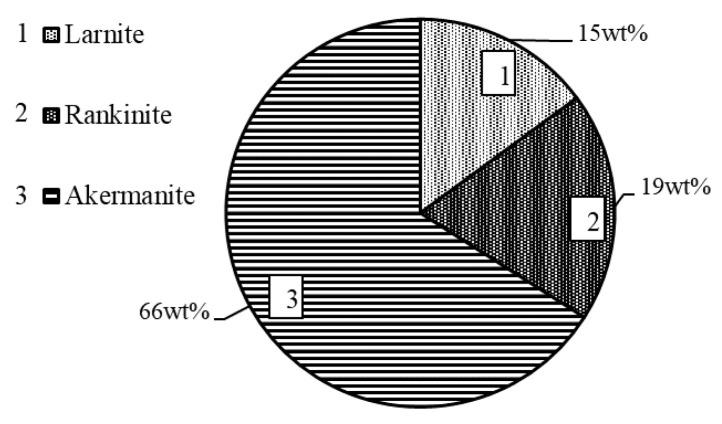
Quantitative composition of the synthesis product at 1150 °C in 45 min with CaO/SiO_2_ = 1.5.

**Figure 11 materials-16-02930-f011:**
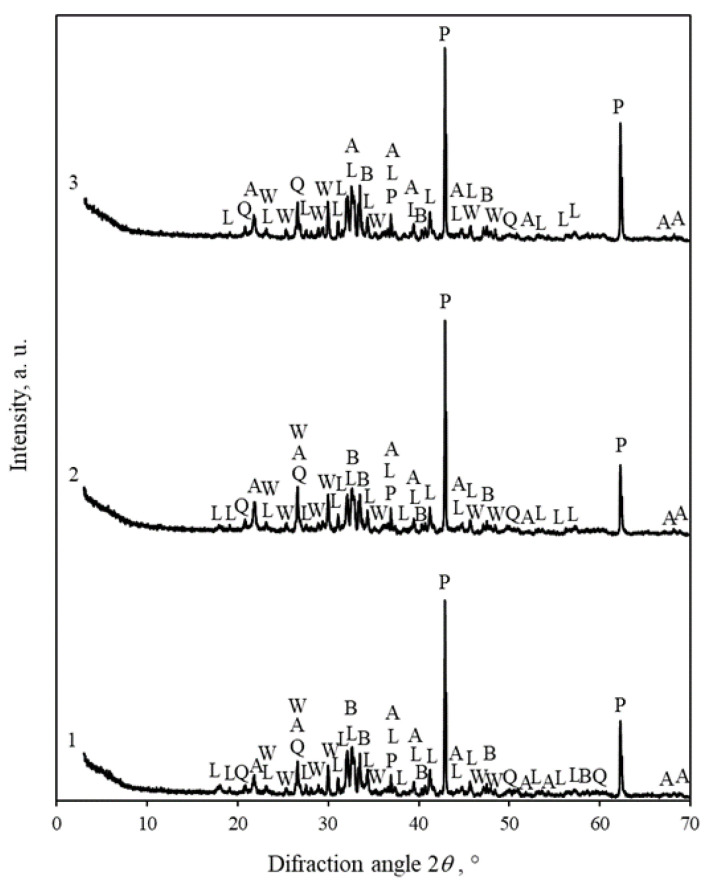
XRD patterns of akermanite synthesized for 1 h (1) and after 24 h of carbonization at 25 °C (2) and 45 °C (3). Indexes: A—akermanite, Q—quartz, W—wollastonite, L—larnite, B—bredigite, P—periclase.

**Figure 12 materials-16-02930-f012:**
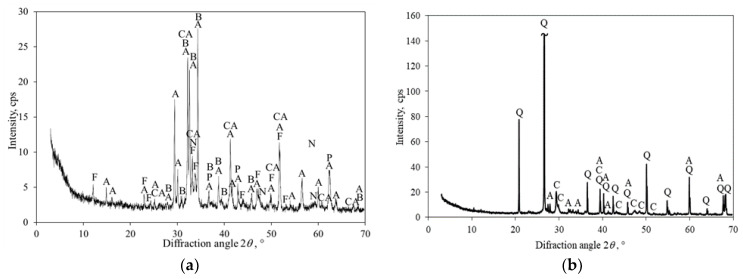
XRD patterns of JSC *Akmenes cementas* OPC clinker (**a**) and after 24 h of carbonization of samples at 45 °C (**b**). Indexes: A—alite, B—belite, CA—tricalcium alumina, F—calcium aluminum iron oxide, P—periclase, N—sodium calcium aluminum oxide, C—calcite, Q—quartz.

**Table 1 materials-16-02930-t001:** Oxide composition of Karpenai limestone, Saltiskiai clay and JSC *Akmenes cementas* clinker, wt%.

**Material**	SiO_2_	CaO	Al_2_O_3_	K_2_O	Na_2_O	MgO	Fe_2_O_3_	SO_3_	Other	LOI
**Limestone**	2.02	47.95	0.71	0.17	–	3.94	1.15	0.23	0.57	43.26
**Clay**	45.64	10.97	14.0	3.06	0.17	4.0	7.04	–	0.22	14.9
**Clinker**	20.69	64.06	5.13	1.12	0.10	3.31	3.25	0.53	CaO_free_ = 1.81	

**Table 2 materials-16-02930-t002:** The compounds, which were identified in the synthesis product, from a mixture of Karpenai limestone and Saltiskiai clay (CaO/SiO_2_ = 1.5) in 25–1150 °C temperature range.

No.	Temperature, °C	Identified Compounds	Remarks
1	25–625	Calcite, quartz, dolomite, illite	
2	650–675	Quartz, calcite, illite	
3	700–750	Akermanite, calcite, quartz	Low intensity peak of akermanite
4	775–900	Akermanite, quartz	High intensity peak of akermanite
5	925	Akermanite, quartz, larnite	Traces of larnite, high intensity peak of akermanite
6	950–1075	Akermanite, quartz, larnite	High intensity peak of akermanite
7	1100	Akermanite, quartz, larnite, rankinite	Traces of rankinite, high intensity peak of akermanite
8	1125–1150	Akermanite, quartz, larnite, rankinite	High intensity peak of akermanite

**Table 3 materials-16-02930-t003:** Compressive strength of the samples formed from the Karpenai limestone and Saltiskiai clay (calcination—1150 °C, 45 min) or OPC clinker mixtures with standard sand (mass ratio 1:3).

No.	Curing Temperature, °C	Binder Synthesized at a Temperature of 1150 °C	OPC Clinker
		Compressive Strength, MPa	
1	25	9.13	53.03
2	45	14.45	67.5

## Data Availability

The data that support the findings of this study are available from the corresponding author, R.S., upon reasonable request.

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
