# Peer review of "The Influence of Mg-Impurities in Raw Materials on the Synthesis of Rankinite Clinker and the Strength of Mortar Hardening in CO2 Environment"

_materials, 2023, doi:10.3390/ma16072930_

Round 1

Reviewer 1 Report

The authors present very interesting experimental results on study the Mg content effect on the production of low-lime binders that harden in CO2 environment. The paper is written very well, all analytical methods are appropriate and the obtained results are explained and discussed. My comments:

1. page 4, line 153 – the abbreviation of differential scanning calorimetry should be corrected to DSC.

2. Analytical equipment (XRD, DSC, etc.) should be reported before presenting the obtained results, e.g. Fig. 1 and Fig. 3.

3. Fig. 4 - All abbreviations should be explained, e.g. Fig. 4 uses formulae/abbreviated notations for chemical compounds in the field of mortar chemistry. In my opinion, not every reader of journal Materials is an expert in the subject area and it would make it easier for the general reader to understand the text.

Author Response

We would like to express our great appreciation to you for all the work done and for the valuable remarks concerning the paper. Your comments and suggestions are very helpful in improving the quality of the article and we have tried to take all of them into consideration.

Reviewer 2 Report

The manuscript deals with the interesting subject of alternative binders, CO2 mitigation and capture. A short description of the CO2 curing process would be useful for the non-expert reader. Is it performed in a chamber? Does it proceed under atmospheric CO2 concentration?

It is shown that rankinite formation is inhibited by the presence of MgO in the raw materials, leading to akermanite formation, a crystalline material that does not react with CO2 and, therefore, is not suitable for CO2 curing low-lime binders. It would have been important to test this hypothesis using a MgO-free raw-material in the same experimental conditions, to check if rankinite would then be formed. Without this check, it is arguable if MgO is the reason why rankinite is absent.

All chemical concentration in weight have to be indicated as ‘wt %’ for clarity sake, as molar proportion could be indicated in % as well.

I am not a native speaker of English myself. Nevertheless, in my opinion, the text needs a review from a native speaker of a professional reviewer.

Abstract

a) it is not clear, why does rankinite production requires “...almost twice as much carbonate rock…”, given that it has a lower CaO molar proportion than the OPC main phases (C3S, C2S)

b) MgO instead of MgO2

c) the Rietveld method is a method for structural analysis of crystalline materials, and not of its chemical composition. Therefore, the following statement is not accurate “According to Rietveld refinement, about half of the Mg2+ ions in the crystal lattice structure of this compound are replaced by Al3+ and Fe3+ ions, and the chemical formula of the resulting compound is Ca2(MgO0.495·FeO0.202·AlO0.303)·(FeO0.248·AlO·Si1.536·O7).”

Introduction

d) the introduction is clear and informative, but the paragraph mentioning ‘1.13 nm tobermorite’ is rather abrupt (line 63) – it may be improved by giving its context or relevance for the present study

e) the term “natural melt rocks” is not correct; the original paper cited [2] uses the term ‘paralava’, indicating a rock produced by natural melting of a sedimentary rock

f) it sounds like advertisement “...the low CO2 emission products offered by Solidia Technologies (USA) are probably the best developed in the world today.

g) CO is used in line 100 – is it CO or CO2?

h) the non-silicate Ca compounds mentioned in the following sentence should be listed  – “...Another way is to replace the calcium silicates with other materials in CO2 hardening concretes at least partially...”

i) the heat source for the calcination process with ‘no- fossil bound CO2 release’ has to be indicated

Materials and methods

j) the statement “The amount of MgO in the composition of another carbonate, dolomite, is moderate, at almost 4%” has to be rewritten, because the amount of MgO in the composition of dolomite is 22 wt%; the “4%” mentioned is the MgO concentration in the whole rock

k) the meaning of ‘dispersive carbonate’ is not clear in the sentence “Taking into account the fact that, during calcination, due to CO2 release, the mass of limestone will decrease by almost two times and the particles will shrink, this type of dispersive limestone is suitable for high temperature synthesis of calcium silicates.”

l) the presence of muscovite (XRD) is mentioned in line 176, while in diffractogram (figure 3) the peak is interpreted as illite

m) illite instead of ilite

Results and discussion

n) in Table 2, instead of “intensive peak...”, use “high intensity peak” on opposition to “low intensity peak”

p) in Table 2, rankinite instead of rankinkte

q) Figure 6 is not clearly showing the begining and and of phase transformations; try a view from the top, using colors of a heatmap

r) details should be given on the Rietveld refinement (software, conditions, steps, Rwp, chi2)

s) in Figure 9, the refinement may be improved in the main peak, around 31 degrees 2-theta; try preferred orientation

t) the procedure in line 359, based on the area of a selected XRD-peak should be illustrated, as peak overlaping is a very likely situation in cement XRD analysis

Author Response

(The authors gave the same response as above.)

Reviewer 3 Report

Siauciunas et al studied The Influence of Mg-impurities in Raw Materials on the Synthesis of Rankinite and the Strength of Mortar Hardening in CO2 Environment. The manuscript has certain innovations but there are problems in the following aspects that need to be revised:

1 The language of the manuscript needs to be greatly improved.

2 All abbreviations in the manuscript should be consistent.

3 The choice of key words is not appropriate enough.

4 The Introduction section is not attractive, it is recommended that the author adjust the layout of this section.

5 The resolution of the pictures in the manuscript is not high enough, it is recommended that the author improve it to a certain extent.

6 The XRD experimental results in Figure 12 should be analyzed in more detail.

Author Response

(The authors gave the same response as above.)

Reviewer 4 Report

For reviewing the manuscript “materials-2279127” intitulled “ The Influence of Mg-impurities in Raw Materials on the Synthesis of Rankinite and the Strength of Mortar Hardening in CO2 Environment” and written by  R. Siauciunas, E. Prichockiene, Z. Valancius. The work is very interesting in point of view environment impact. The authors propose clinker manufacturing with temperatures less than 1300°C. There are interesting results. However, when we read this paper we feel lost where the structuration of this paper pose problem for understanding. For this reason, I suggest to the authors the restructuration of this paper.

 Title 

When we read the title, we wait to find article talking about synthetized Rankinite structure. However when we read this paper we understand that we are manufacturing clinker with Rankinite structure which is completely different. As I had and I have chance to work about clinker and reduce the temperature of clinkerisation, I had chance to work with pure phases (C3S, C2S, C3A, C4AF, C3S2,….). For example if we want to talk about synthetized  Rankinite structure we can for example use CSH and do the calcination or use source of CaO and source of SiO2 (of course we should respect ratio C/S) in this we can talk only about Rankinite. However, in this paper the authors have used the same raw materials than standard clinker (which is interesting) where they have just take in account ratio C/S less to 2 but we are always with clinker manufacturing. For this reason, I suggest to add clinker to the title and Rankinite in order to show we talking about Rankinite in clinker. I will not give proposition of title in order to let choice for the authors. It is their work and I prefer to give this this choice. I am here only to help them for improving their interesting work. 

Abstract

Your abstract should be rewritten. You should put oand talk only about your work. The first paragraph should be removed. In abstract you can put on sentence about context, after the aim of your work (that you have manufactured clinker with different ratio of C/S), the characterization techniques and pertinent results with values if possible. This is what we should find in your abstract. When we read it we understand what you have done. With your abstract, we feel lost. 

Introduction

For the introduction, I propose to the authors to bring the introduction of the abstract will be added in this introduction as the first paragraph in order to introduce that you are interested to the clinker. After you go to the Rankinite structure and let your introduction, which is well done.

Materials and methods

The part of Materials and methods should be rewritten in order to give more visibility. The authors have given lot of information and results without any exploitation. When we read your Materials and methods we feel lost. For this reason I suggest:

First we beguine with Materials:

Materials

Your text  

“ JSC Akmenes Cementas has been successfully producing Ordinary Portland cement (OPC) from local raw materials (Karpenai deposit limestone and Saltiskiai deposit clay, Lithuania) for more than 7 decades. Its main constituents are also anhydrous calcium silicates – alite (3CaO·SiO2; ~66%) and belite (2CaO·SiO2; ~10%). The only significant difference is that the molar ratio CaO/SiO2 of the raw material mixture is by one and a half points higher (2.7) than that required for the synthesis of rankinite (1.5). It is likely that these raw materials are also suitable for the production of low-lime binders which would harden in CO2 environment. For comparison, we also determined the compressive strength and the mineralogical composition of mortar samples formed from OPC clinker produced by JSC Akmenes Cementas/sand mixture and hardened in CO2 environment”

Should be removed from materials section. In Materials part you should talk only the materials used;

Limestone: Provenance Company, chemical composition, volumic mass and Blaine surface area (if you give Blaine surface area, it means you have measured volumic mass), …..

Clays:    Provenance Company, chemical composition, volumic mass and Blaine surface area (if you give Blaine surface area, it means you have measured volumic mass), …..

Conventional Clinker:  Provenance Company, chemical composition, volumic mass and Blaine surface area (if you give Blaine surface area, it means you have measured volumic mass), …..

You put one or two tables with all information: chemical compostion, Baline SA, VM,

For granulo (PSD) if you have possibility to add calcined clinker that you have prepared, please add all in results and discussion for all PSD and discuss difference between your clinkers and conventional clinker (if you have monodal or bimodal distribution). Please give one figure with volume versus particle size (diameter) and cumulative volume (passing) with log particle size (use two X axes) and you add D10, D50 and D90 of your clinker and raw materials in order to show if you are near D50 (10-15 µm) for clinkers it means you are proch to CEMI generally. Also for limestone (D50  5-10µm, means no agglomeration) and clay (D50 4 to 5µm means no agglomeration). All these information allow the understanding heat behaviour.

I say if possible it means if you have not these results you let PDS in materials and method just you put it in one figure and you put with log for passing. Be careful please your figures have problems with X axe try to trace correct abscise axes    

Please for XRD patterns and TGA you put them in results and discussion (please see my recommendations in results and discussion).

After Materials you go the methods

Methods

Preparation methods (how you have prepared all from raw materials to mortar)

If possible, to add one or two schematic figures showing your procedure means from raw materials to the mortar confection. In methods you will start with raw materials how you prepare it (heating, milling, sieving,…..). after you explain how you have prepared your clinkers by different ratio and how is it calcination……..

After you go to the mortar and you should give more details (ratio water, sand is it normalized it means 0-2 mm,…………) and why you have chosen to cure your samples in CO2 environment what you look for.  Your paper is proposed to all readers and believe me in cement researchers they don’t know that there is patents in CO2 injection.

Just I precise that is suggestion for improving your paper; however, you can use your terms which you consider more appropriate for you. You are not obliged to use my title (Preparation methods)

Second part is

Characterization techniques   

In this part, the authors should give more details about equipment’s and machines used

For example:

     X-ray fluorescence spectrometry, S8 that is technique of WDS with more than 7 mono-crystal and very well precision. However, you should show how you have found your chemical composition is it using fused bead where you should give details. Or you have used Samples in compacted pellets containing wax for x-ray fluorescence. These tow technique are very recommended in cement companies because we get good precision and we use vacuum only. However, you can use also directly powders where you will use helium gas not vacuum. You should precise which technique that you have used. It is important these information for readers.

Now I go to the X-Ray diffraction. I precise to the authors that you have used good equipment’s. Like I have chance, to work with several cement companies in Europe and I confirm that most of them use Bruker mark. Concening D8, you should give more details about angle domain, increments taken and especially, which wavelength you have used is it copper or Cobalt in your case I think copper with 1.5434 A° you should precise all that. Now how about heating for XRD in situ have you chamber furnace or you have added furnace. Which heat rate you have used. Give more details please.

Now I go to the software: For Eva which ICDD library have you used is it PDF2 or PDF4. If you have used PDF2 it means that you cannot get STR file which are used for TOPAS. If you have PDF4 you can create your STR file used by TOPAS. All these details are important.

When I see your text

“In order to determine the mineral quantitative composition, the research was supplemented with the Rietveld refinement method (10% ZnO additive was mixed into the samples, and Diffrac.Topas 4.2 software was used)”  

When I read this text, I say that the authors are not putting in value this interesting work.

TOPAS is software based in the Rietveld refinement method allowing the crystalline phase quantification composition. However Why the authors add ZnO????????

You should know that this technique is used by most of European cement companies (in china and india we can find others techniques). In TOPAS you can standardization (standard method) technique (intern and extern). In intern method, is adding good crystalline addition to our samples in order to get good based line for getting good amorphous quantification (and crystalline phases quantification of cours). ZnO has good crystalline phase, which allow us to have good-based line (without any reactions) for calculation of amorphous percentage. Generally, we add from 10% to 20%. Please take time to add the details which you show the add value of your work.

Try to give more details about TGA and others machines please.

Results and discussion

The authors studied Rankinite phase and with this structure, we can obtain it by different cases. As shown in difference references cited by the authors, the easy way to get this structure is to go with binary diagram it means we play with only CaO element and SiO2 element we can globally control different parameters.

For this, with different references that the authors have cited I suggest these references in order to have an idea (just for reading you are not obliged to cite them).

Mats Hillert et al,  CALHlAD Vol. 15, No. 1, pi. 53-58, 1991.

 Warda Ashraf et al Cement and Concrete Research 100 (2017) 361–372

However, when you with ternary elements (as in clinker) CaO, SiO2 and Al2O3 here we are in complex situation. In addition generally when we want to prepare clinker respecting the personages of C3S (60-64%), C2S (15-20%),……. We should prepare good raw clinker (mixture before clinkerization or calcination) by doing rectifications with meniral additions in order to get good index parameters that cement companies used [empirical parameters as LSF (Lime Saturation Factor), SM,…please read this paper for understanding: Bouchenafa et al Constr. Mater. 2022, 2, 200–216. https://doi.org/10.3390/constrmater2040014]. All that can affect the formation of specific structure of C/S less than 2. This introduction is in order to show to the authors that your study is very interesting and more complex and thank you for your work.

Now in order to improve this paper I give these sugesttions:

In results and discussions, you will propose three parts with interpretation and discussion with comparison to the literature. Please use your well references of your introduction for discuss your results.

The first concerning

Raw materials used

 You take XRD and TGA from materials and methods (if possible all granulo PSD of clinkers) and put it here in this part and you do depth discussion. You should show the specificities of your raw materials: as your limestone there is the dolomite and quartz (it means your calcite is not pur) and also your clay you have kaolinite which is T-O structure, illite which is T-O-T structure and Montmorilinite which is T-O-T structure. All these information are well to show them and compere with literature.

The second part concern

Clinker formation

Here you will put Table 2 figure 6, 7, 8 and 9. First, you have used different conditions without any explications why these conditions are chosen. For example why temperatures of 1100, 1150 and 1200, why 30 min, 45 min and 60 min. After XRD in situ you can justify that after 1100 °C we got all structures. You can give references for your selections of these conditions.

This part should interpreted and discussed with comparison to literature please use good references that you have cited in your introduction.

After you go to the Mortar samples (with carbonation) with the rest of your figures and you do good discussion with literature.

Conclusion

Conclusion should be presented as function of your new version.

Conclusion

Major revision and thank you for your work and I hope that you will improve this paper.

You can ask editor more time          

Best regards           

Author Response

(The authors gave the same response as above.)

Round 2

Reviewer 2 Report

Dear authors, 

the detailed answers you provide for each topic of my review and respective adjustments in the manuscript are correct and complete. The weaknesses I noticed in the first version were mostly problems of language or shortcuts that hindered a full comprehension of the paper.

I recommend the manuscript for publication in the present form.

regards, 

Reviewer 3 Report

Accept in present form

Reviewer 4 Report

Thanks to the authors for their effort and I am sorry if I have given lot of recommendations for short time. I accept it